# Interaction between Cd and Zn on Metal Accumulation, Translocation and Mineral Nutrition in Tall Fescue (*Festuca arundinacea*)

**DOI:** 10.3390/ijms20133332

**Published:** 2019-07-06

**Authors:** Qin Dong, Shuai Hu, Ling Fei, Lijiao Liu, Zhaolong Wang

**Affiliations:** School of Agriculture and Biology, Shanghai Jiaotong University, Shanghai 200240, China

**Keywords:** phytoremediation, Zn/Cd interaction, mineral nutrition, Cd translocation

## Abstract

Tall fescue (*Festuca arundinacea*), an accumulator that is able to accumulate and excrete cadmium (Cd), has attracted much attention for its possible use in phytoremediation of heavy metal contaminated soils. In the present study, the interaction between Cd and Zn, and their uptake, translocation and accumulation under external Cd and Zn treatment in tall fescue were investigated. The concentrations of K, Ca, Mg in xylem sap under Cd and Zn treatment were measured to determine the level of mineral nutrients and their relationship with Cd alleviation. The result showed that Cd and Zn antagonized each other in the roots, while Cd antagonized Zn and Zn synergized Cd in the shoots of tall fescue. Compared with Cd only treatment, the concentrations of Ca, Mg and K in xylem sap increased after the addition of Zn, and they increased the most in the guttation. This result indicated that the addition of Zn facilitates the level of mineral elements to alleviate Cd toxicity, which might be used to improve the phytoremediation efficiency of Cd contaminated soils by tall fescue.

## 1. Introduction

Cadmium (Cd), as a non-essential element of life, is one of the major environmental threats in agricultural systems due to its strong biological toxicity and mobility [1,2]. Serious environmental pollution can be caused by the large amounts of cadmium in water and soil, and it is easily absorbed and accumulated by plants, then endangering human health through the food chain [3]. Phytoremediation is considered to be a better remediation technique of contaminated soil compared to physical remediation and chemical remediation because of its advantages of low cost, no damage to soil structure and no secondary pollution [4]. However, phytoremediation is also limited by the problems of small biomass and long remediation period of hyperaccumulators. 

Zinc (Zn) is an essential micronutrient for plants, which has a high chemical similarity with Cd. Zn and Cd are usually co-located in the soil, resulting in their ability to easily interact with each other both in the soil and root absorption, translocation, and accumulation in plants. There were many studies on the interaction between Zn and Cd [5,6]. Zn and Cd can antagonize each other in plant roots and compete for their co-transporters. In *Brassica napus*, mixed treatment of Cd + Zn decreased shoot accumulations of Zn and Cd when compared to Cd or Zn treatment alone [7]. Li [5] reported that Zn enhanced Cd accumulation in shoots of the Cd/Zn hyperaccumulating ecotype of *Sedum alfredii*. However, Zn synergized with Cd, and Cd antagonized Zn in *Potentilla griffithii* [8]. The inconsistent results of interaction between Zn and Cd could be from different plant species or ecotypes and the concentrations of Zn and Cd treatment [7,8,9]. 

Cd is very toxic to plants, which could inhibit plant biochemical and physiological processes, root uptake of the mineral nutrients [10,11,12,13], and even cause plant death [14]. Some studies showed that nutrient elements could play a protective role when plants are stressed by Cd [15,16]. Additions of K, Ca, Mg, and Zn reduced Cd uptake and alleviated the Cd stress by inhibiting the root uptake or enhancing the antioxidant ability [17,18,19,20]. 

Tall fescue (*Festuca arundinacea*) can uptake, translocate and accumulate more Cd than Cd hyperaccumulator, *Solanum nigrum*, and with the advantages of large biomass, it can be an excellent choice for phytoremediation [21]. Our pre-study showed that Zn interacted with Cd absorption and accumulation in tall fescue. Therefore, the objective of this study was to investigate Zn and Cd interactions on plant uptake, root–shoot translocation and shoot accumulation in tall fescue.

## 2. Results

### 2.1. Plant Growth

No significant differences of plant growth were noticed between control and Zn treatment (no Cd) as well as Cd and Cd + Zn treatments (Cd stress), indicating that Zn excess application did not result in significant impact on plant growth as indicated by plant root length, shoot biomass, and root biomass (Figure 1). Only 10.3% of plant height reduction was observed by Zn treatment compared to control. Cd stress (75 μM Cd) resulted in significant negative impacts on plant height, root length, shoot biomass, and root biomass. Cd significantly decreased the plant height by 32.0% and 28.9%, root length by 32.2% and 34.8%, root biomass by 29.0% and 40.3%, under 15 μM (Cd treatment vs. control) and 115 μM (Cd + Zn treatment vs. Zn treatment) of Zn treatments, respectively. Cd did not show significant impact on shoot biomass under 15 μM Zn treatment, but 16.2% of shoot biomass reduction was observed under 115 μM Zn treatment.

### 2.2. The Uptake, Accumulation and Translocation of Cd

Under Cd treatment, tall fescue accumulated as high as 4468.9 mg/kg of Cd concentration in roots and 130.0 mg/kg of Cd concentration in shoots at normal Zn nutrient level (15 μM) (Figure 2). Excess addition of Zn (115 μM) significantly decreased the Cd concentration in roots by 68.3%, but increased the Cd concentration in shoots by 69.1%, which resulted a 203.1% increase of Cd translocation factor, indicating that Zn could facilitate Cd translocation from roots to shoots.

### 2.3. The Uptake, Accumulation and Translocation of Zn

Excess addition of Zn resulted in a significant increase of Zn concentration in roots and shoots, but a decrease of Zn translocation (Figure 3). Zn and Cd + Zn treatments significantly enhanced the root Zn concentration by 534.5% and 202.9%, and shoot Zn concentration by 165.9% and 137.6% compared to control and Cd treatments, respectively. In contrast, increasing solution Zn concentration decreased the Zn translocation factor by 57.2% with the absence of Cd (Zn treatment vs. control), while no significant difference between them was observed under 75 μM Cd treatment (Cd + Zn treatment vs. Cd treatment). Cd stress (75 μM) did not show significant impact on Zn concentration in roots at 15 μM Zn (control vs. Cd treatment), but a 32.3% reduction was observed at 115 μM Zn (Cd + Zn treatment vs. Zn treatment). The same negative impact was noticed on shoot Zn concentration and Zn translocation factor. Cd treatment significantly decreased shoot Zn concentration by 56.7% and 61.3%, and Zn translocation factor by 69.1% and 44.6% under 15 μM and 115 μM Zn treatments, respectively. These results could be explained by the significant interaction between Cd and Zn (Table 1).

### 2.4. Amount of Cd and Zn Accumulated in Tall Fescue

Under 75 μM Cd stress, increasing Zn concentration decreased Cd amount in the whole tall fescue plant by 42.3% (Figure 4a). However, Zn amount was increased by 322.8% under Zn treatment compared to control and 145.1% under Cd + Zn treatment compared to Cd treatment (Figure 4b). It is obvious that the magnitude of the increase was greater at 0 μM Cd than that at 75 μM Cd, indicating that Cd addition could inhibit the accumulation of Zn in tall fescue. Under 115 μM Zn treatment, the total Zn amount significantly decreased by 64.6% at 75 μM Cd compared to the control (0 μM Cd), while the difference was not significant under 15 μM Zn treatment. Significant interaction between Cd and Zn was observed on Zn amount (Table 1).

### 2.5. Interaction between Zn and Cd on Plant Growth and Metal Concentrations 

As is shown in Table 1, according to the analysis of Cd and Zn treatment effects and their interactions on plant growth, it is obvious that the effects of Cd were significant, while those of Zn were only significant on the plant height, and the interactions between Cd and Zn were not significant on plant height, root length, shoot and root biomass. For root and shoot Zn concentration, Zn translocation and Zn amount, the effects of Cd and Zn treatment and their interactions were all significant (except the effect of Cd on root Zn concentration). Cd and Zn treatment effects and their interactions on Zn, Ca, Mg and K concentrations in xylem sap were significant for the most part, except the effects of Cd and Cd/Zn interaction on Zn concentration in xylem sap of sheath, Zn and Cd/Zn interaction on Mg concentration in xylem sap of sheath, Cd and Zn effects on Mg concentration in xylem sap of leaf blade, and Cd and Cd/Zn interaction on K concentrations in xylem sap of sheath and guttation.

As is shown in Table 2, the concentrations and summation of Cd^2+^ and Zn^2+^ under the four treatments were calculated respectively. The concentration of Cd^2+^ under Cd + Zn treatment decreased almost a half compared to only Cd treatment. The concentrations of Zn^2+^ were at a low level under control and Cd treatment, and increased significantly under Cd + Zn treatment, while the highest level was observed under Zn treatment. The summations of Cd^2+^ and Zn^2+^ were at low level only under the control, and increased to the same high level under Cd, Zn and Cd + Zn treatments.

### 2.6. Mineral Nutrients in Xylem Sap

Figure 5a showed that the Cd concentration in xylem sap of sheath, leaf blade and guttation was in descending order under Cd treatment, and the differences were significant. Cd concentration in guttation was only 16.5% of that in xylem sap of sheath. Cd concentration in the three positions of xylem sap under Zn treatment increased 34.8%, 51.8% and 87.7%, respectively. The difference between them was not significant and the highest increasing range occurred in guttation. For Zn concentration in xylem sap, no gradient was observed between different positions under control, Cd treatment and Cd + Zn treatment (Figure 5b). Cd only treatment resulted in no impact on Zn concentration in xylem sap at any position. External Zn significantly enhanced it by 12.2 times at sheath, 17.8 times at leaf blade and 160.5 times at guttation. The addition of Zn and Cd in combination enhanced it further at sheath and leaf blade; however, a 62.0% decrease was observed at guttation compared to Zn only treatment.

For control, Zn and Cd only treatment, there was no significant gradient of Ca concentration in xylem sap between different positions, but Mg and K concentrations decreased significantly from bottom to top in the three positions of xylem sap (except Mg concentration under Zn only treatment). When Cd and Zn were added mixedly, the concentrations of Ca, Mg and K in xylem sap of all positions increased significantly. Compared to the control, Ca concentration under Zn + Cd treatment increased 178.4%, 171.0% and 305.9%, Mg concentration increased 55.4%, 58.9% and 327.6%, K concentration increased 49.8%, 745.2% and 1225.1%, at sheath, leaf blade and guttation, respectively. It is obvious that the magnitude of the increase in guttation was the greatest, indicating that the addition of Zn and Cd in combination could enhance the translocation of Ca, Mg and K in tall fescue shoots.

## 3. Discussion

### 3.1. The Effect of Cd and Zn on Plant Growth

In this study, the growth of tall fescue was affected in different degrees by the change of Zn and Cd concentrations in nutrient solution and no significant interaction between Zn and Cd was observed. The growth of roots were inhibited significantly by Cd, while the addition of Zn had an effect on it (Figure 1c,d), which was consistent with the results of Benakova [9]. However, it must be emphasized that the toxicity of Zn depends on the concentration added, and excessive Zn can inhibit the growth of plant roots [22,23,24]. The response of shoot growth to Zn and Cd treatments was different from that of root system. For plant height, Zn and Cd showed antagonistic effects on each other, while both of them had no effect on shoot biomass (Figure 1a,b). This is contrary to results of Ozturk [25]. The reasons may be the different concentrations of Zn and Cd in nutrient solution, the different culture methods or the different sensitivity of plants to Zn and Cd. During the experiment, it was obvious that the concentration of Zn and Cd in the new leaves was significantly lower than that in the old leaves after heavy metal treatment. Additionally, the growth rate of new leaves of tall fescue was faster, while the length of each leaf was shorter, so that the plant maintained a faster metabolism rate. This mechanism resulted in the fact that the plant height of tall fescue treated with heavy metals was lower than that of the control but the shoot biomass remained unchanged, which also ensured that tall fescue could maintain normal growth and physiological and biochemical processes under a certain concentration of heavy metals.

### 3.2. The Interaction between Cd and Zn in Roots of Tall Fescue

After 15 days of Cd treatment, the Cd concentration in the roots of tall fescue decreased by more than a half with the increase of Zn concentration in the nutrient solution (Figure 2a), while Zn concentration in the roots showed a decreasing trend with the increase of Cd concentration, indicating that Zn and Cd acted antagonistically to inhibit their uptake in roots of tall fescue. This is consistent with the results of Cheng [6]. Increasing Zn concentration in nutrient solution decreased Cd concentration in the root of *Carpobrotus rossii* by 39–66%, and the addition of Cd decreased Zn concentration in the root by more than 50% under 115 μM Zn treatment. The reason may be that some transporters in the root system, such as HMA2, HMA3, ZIP family and NRAMP family can simultaneously transport Zn, Cd and other ions [26,27,28,29]. As is shown in Table 1, the concentrations and summation of Cd^2+^ and Zn^2+^ under the four treatments were calculated respectively. It was found that the absorption capacity of the roots was not saturated without the external Cd^2+^ and Zn^2+^. When adding Cd^2+^ or Zn^2+^ alone or in combination to the nutrient solution, the sum of the uptake capacity of Cd^2+^ and Zn^2+^ was stable. In the subsequent experiments, the impact of Zn on the absorption of Cd by the root can be eliminated through foliar spraying and other techniques, so as to achieve the purpose of absorbing more Cd.

### 3.3. The Interaction between Cd and Zn in Shoots of Tall Fescue

Once it enters the root cells, Cd is transported to the shoots through the xylem sap and accumulates in the shoots. During this process, the mechanism of interaction between Cd and Zn was different from that in roots. As a whole, Cd concentration in the shoots of tall fescue increased significantly after the addition of Zn; because of the decrease of Cd concentration in the roots, Cd translocation factor was increased, while the amount of Cd was decreased. Zn concentrations in the shoots and roots both decreased after the addition of Cd, so Zn translocation factor was decreased and the amount of Zn showed a downward trend, too. The possible reasons are as follows:

First, the cadmium toxicity leads to rapid synthesis of PC thiol-based complexing substances [30]. Phytochelatins can form Cd–PC complexes, and Cd is sequestrated in the vacuole of root cells, thereby reducing the translocation of Cd from roots to shoots [31]. However, in the presence of Zn, the formation of Zn–PC complexes can increase the concentration of free Cd, thus enhancing the translocation of Cd from roots to shoots [32]. Therefore, Cd in roots of tall fescue could translocate to the shoots more effectively by the addition of Zn, which is conducive to promoting phytoextraction. 

The second reason is the change of xylem loading capacity caused by the competition between Zn and Cd. The translocation of Zn and Cd from the roots to shoots requires active loading to xylem [33]. In this research, the capacity of xylem to load Zn and Cd is related to the concentration of Zn and Cd in the shoots. The chemical similarity between Zn and Cd leads to the affinity of Cd to the binding site of Zn transporter, so Zn transporters ZNT1, ZNT5 and MTP1 should have the ability to bind and transport Cd [34]. In Kupper and Kochian’s [34] research, the addition of Cd significantly reduced the Zn translocation to the shoots of Ganges ecotype of Cd/Zn hyperaccumulator *T. caerulescens*. Meanwhile, Zn deficiency in mesophyll caused by elevated Cd was also involved in the down-regulation of ZNT1 expression in *T. caerulescens*. Therefore, the addition of Cd inhibited the xylem loading of Zn, leading to a large amount of Cd^2+^ entering the xylem. This may be one of the possible reasons for the decrease of Zn translocation under elevated Cd. In the study of *Potentilla griffithii*, the antagonism or synergy interaction between Cd and Zn suggests that the two cations enter the plant through a common transport system [8]. In addition, Zn transporters may have more affinity for Cd in tall fescue shoots, so the addition of Cd to plants treated with Zn reduced the Zn translocation due to the competition and interaction between these metals. Since Cd utilizes other transporters, including Zn transporters, for the uptake and translocation, the addition of Zn to plants treated with Cd could facilitate the translocation of Cd.

The third reason is that the addition of Zn promotes the translocation of the mineral nutrients with xylem sap and transpiration, thus increasing the mobility of Cd from roots to shoots. Most of the ions taken up by the roots were translocated to the shoots through transpiration and distributed to each leaf with the xylem sap [35,36,37]. Because of the characteristic of tall fescue leaf structure, the xylem sap finally excretes out of the plant through the terminal of vessel, the hydathode [38]. The determination of ion concentrations in xylem sap of different positions can reflect the translocation and distribution of ions in plant shoots effectively and directly. After the addition of Zn in the tall fescue treated with Cd, it was obvious that Cd concentration in the xylem sap of sheath, leaf blade and guttation all increased significantly, which is the reason why shoot Cd concentration and Cd translocation factor both showed an increasing trend. Cd and other mineral nutrients are translocated from roots to shoots through xylem vessel by xylem flow and are therefore influenced by root pressure and transpiration. The stronger the transpiration, the faster and greater the xylem sap is transported to the shoots. In Stark’s [39] research, transpiration is inhibited in the absence of nutrients. In this experiment, the concentrations of Ca, Mg and K in xylem sap increased after adding Zn, which was consistent with the research of Stoyanova and Doncheva [24], Jiang [40] and Benakova [9]. On the one hand, mineral nutrients can improve the physiological function of plants, thus improving the resistance of plants. For instance, Ca is an important component of cell wall, and it can also maintain the osmotic balance of nutrients in plants and the integrity and permeability of protoplasmic membrane; Mg is an important component of chlorophyll molecule and is directly related to photosynthesis [41]; K can promote the activation of enzyme system and photosynthesis, regulate the transpiration, and plays an important role in plant stress resistance, such as drought and cold resistance [42,43]. On the other hand, increased concentrations of mineral nutrients also promote transpiration, creating a virtuous cycle that increases the amount of Cd carried to the shoots by xylem sap. The interaction between Zn and Cd stimulated the concentration of Ca, Mg and K in xylem sap under most circumstance, which can be beneficial to the growth of plant.

Most previous studies about the interaction between Zn and Cd focused on the roots of plants, and the differences of the interactions between roots and shoots were not well understood. In this study, we investigated Zn and Cd interactions on plant uptake, root–shoot translocation and shoot accumulation in tall fescue via measuring the concentration of Zn, Cd and other mineral nutrients in tissues and xylem sap. We found that the Cd uptake of tall fescue roots could be stimulated, and the Cd translocation and accumulation of tall fescue shoots could be inhibited by adding an appropriate amount of Zn to the nutrient solution. According to our analysis, the interaction between Zn and Cd functioned in different ways in roots and shoots.

## 4. Materials and Methods

### 4.1. Plant Materials and Growth Conditions

Hydroponic experiments were conducted to investigate Zn and Cd interactions on plant uptake, root accumulation and root–shoot translocation in tall fescue. Seeds of tall fescue (cv ‘jaguar 4G’) (bought from Clover company, Beijing, China) were sown in sand culture. After seedlings had grown for 60 days, the uniform plants were selected and transplanted to containers containing 2.4 L 1/2 Hoagland’s solution (25 plants per pot). The solution pH was adjusted to 6.5 with 0.1 M NaOH or HCl, as required. The nutrient solution in the growth containers was continuously aerated with pumps and renewed weekly. The pots were placed in a climate chamber maintained at temperature of 25 °C for 14 h day and 20 °C for 10 h night, 400 μmol m^−2^ s^−1^ of PAR and 50 ± 2% of relative humidity.

### 4.2. Treatments and Experiment Design

For this experiment, Cadmium was added as CdCl_2_•2.5H_2_O at 0 μM (control) and 75 μM, and zinc was added as ZnSO_4_ at 15 μM (control) and 115 μM when the plants grew for two weeks. The four treatments were as follows: 0 μM Cd + 15 μM Zn (control), 75 μM Cd + 15 μM Zn (Cd), 0 μM Cd + 115 μM Zn (Zn) and 75 μM Cd + 115 μM Zn (Cd + Zn). During the 15 days’ treatment, the growth conditions were the same as those described above. Plant height, root length, shoot biomass and root biomass were measured at the 15th day, and the concentrations of Cd, Zn, Ca, Mg, and K in xylem sap and guttation of tall fescue were measured. After harvest, concentrations of Cd and Zn in roots and shoots were measured. During the experiments, the hydroponic solutions were changed every 5 days to maintain consistency among treatments. There were 4 replicates for each treatment.

### 4.3. Measurement

#### 4.3.1. Collection and Determination of Ions in Guttation and Xylem Sap

In order to research the distribution and translocation of the mineral elements and the relationship between them and the accumulation and translocation of Cd, the concentration of Cd^2+^, Zn^2+^, Ca^2+^, Mg^2+^ and K^+^ in the xylem sap (from bottom to top: xylem sap of sheath, xylem sap of leaf blade, guttation) of the second upper leaves was measured. 

Guttation fluid from the hydathodes of the leaf tips exuded during the dark period. During the treatment period, guttation fluid of the second upper leaves was collected by microsyringe at the 15th day at 80–90% of relative humidity in the growth chamber during the 10 h night. Xylem sap was collected after guttation collection with the same relative humidity. To collect xylem sap of leaf blades, they were cut crossways at half of the whole blade, and xylem sap of the second upper leaves was collected by microsyringe during the next 10 h night. Then, the sheath of each plant was cut at 2 cm toward the base and xylem sap was collected in the same way. In order to ensure enough guttation fluid and xylem sap for the concentration determination of each ion, 400 plants of each treatment (100 plants for each replicate × 4 replicates) were used to the collection and the volumes of them were recorded. The collected guttation fluid and xylem sap were diluted to 5 mL by deionized water then the concentration of Cd^2+^, Zn^2+^, Ca^2+^, Mg^2+^ and K^+^ in the solution were determined by the inductively coupled plasma optical (ICP Optima 8000, PerkinElmer, Wellesley, MA, USA).

#### 4.3.2. Determination of Cd and Zn Concentration in Shoots and Roots

After 15 days of treatments, plants were harvested and separated into shoots and roots. The second upper leaves were used to determine Cd and Zn concentration in them. The roots were immersed in 20 mM EDTA-Na_2_ solution for 15 min to remove Cd adhered to the root surface [44], washed with deionized water, and then dried with the absorbent paper. The shoots and roots were dried at 105 °C for 30 min, then oven-dried at 80 °C to a constant weight and the dry weight was recorded as the shoot biomass and root biomass. Dried plant samples were crushed, passed through a 100-mesh (0.15 mm) sieve, and digested in supra-pure concentrated HNO_3_ and H_2_O_2_ (2:1, *v*/*v*) at 120 °C [45], diluted to 50 mL. The concentrations of Cd and Zn were determined by ICP as described above.

Cd translocation factor (TF) was calculated as:

TF = Cd concentration in shoots/Cd concentration in roots.

### 4.4. Statistical Analysis

All data are presented as means of three replicated measurements. Statistical analysis was performed with the software SAS (version 9.1, SAS Institute Inc., Carly, North Carolina, USA) using the general linear model (GLM) procedure. Least significant difference (LSD) at a 0.05 probability level was used to detect the differences between treatments.

## 5. Conclusions

Cd and Zn antagonize each other in the roots of tall fescue, while Cd antagonizes Zn and Zn synergizes Cd in the shoots. The addition of Zn can improve Cd translocation from roots to shoots and its accumulation in the shoots without any impact on the growth of tall fescue. Our results indicate that Zn facilitates the level of mineral elements to maintain the growth of the plant, which might be used to improve the phytoremediation efficiency of Cd contaminated soils by tall fescue.

## Figures and Tables

**Figure 1 ijms-20-03332-f001:**
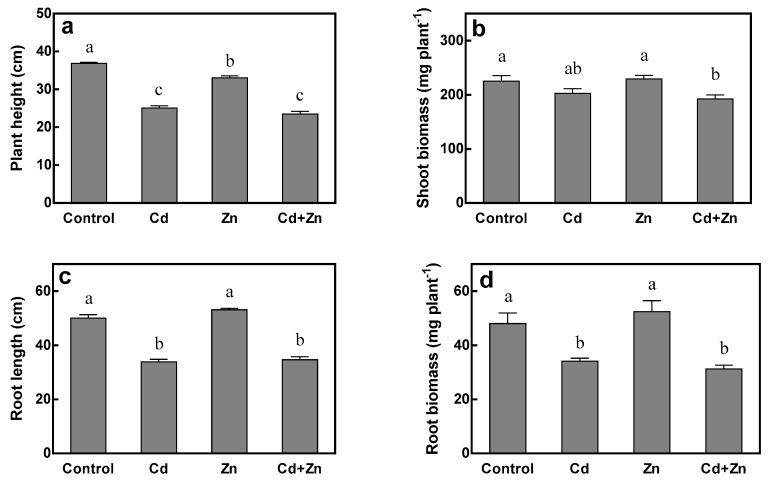
Plant height (**a**), shoot biomass (**b**), root length (**c**) and root biomass (**d**) of tall fescue grown for 15 days in solutions under control (0 μM Cd + 15 μM Zn), Cd (75 μM Cd + 15 μM Zn), Zn (0 μM Cd + 115 μM Zn) and Cd + Zn (75 μM Cd + 115 μM Zn) treatments. Error bars represent ± SEM of three replicates. Means with a common letter did not differ significantly (Duncan’s test, *p* < 0.05). The two-way analysis was performed.

**Figure 2 ijms-20-03332-f002:**
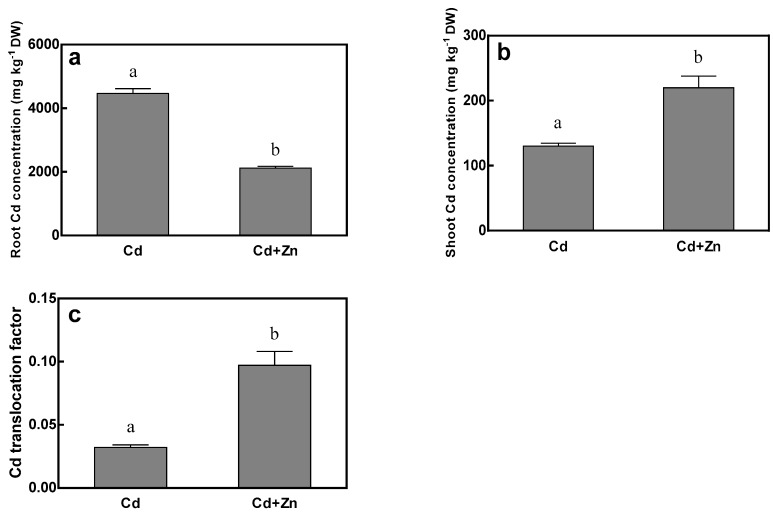
Cd concentrations of root (**a**) and shoot (**b**), and Cd translocation factor (**c**) of tall fescue. Plants were grown for 15 days in solutions under Cd (75 μM Cd + 15 μM Zn) and Cd + Zn (75 μM Cd + 115 μM Zn) treatments. Error bars represent ± SEM of three replicates. Means with a common letter did not differ significantly (Duncan’s test, *p* < 0.05). The two-way analysis was performed.

**Figure 3 ijms-20-03332-f003:**
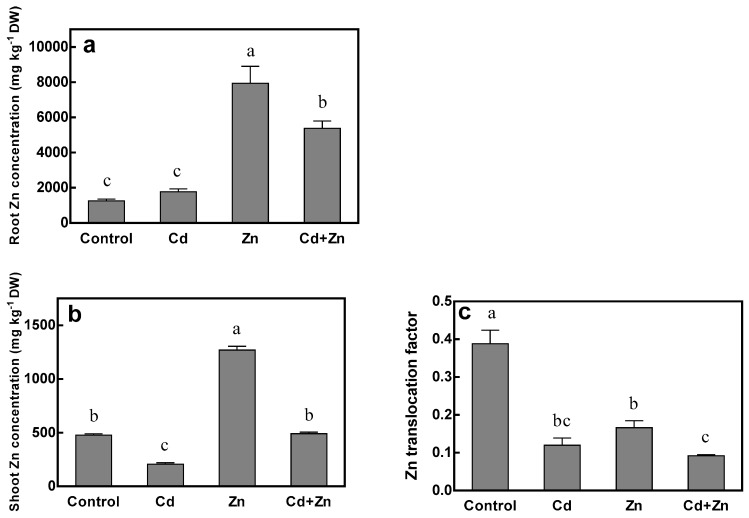
Zn concentrations of root (**a**) and shoot (**b**), and Zn translocation factor (**c**) of tall fescue. Plants were grown for 15 days in solutions under control (0 μM Cd + 15 μM Zn), Cd (75 μM Cd + 15 μM Zn), Zn (0 μM Cd + 115 μM Zn) and Cd + Zn (75 μM Cd + 115 μM Zn) treatments. Error bars represent ± SEM of three replicates. Means with a common letter did not differ significantly (Duncan’s test, *p* < 0.05). The two-way analysis was performed.

**Figure 4 ijms-20-03332-f004:**
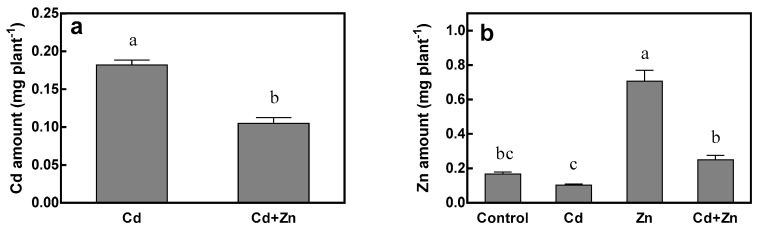
Cd amount (**a**) and Zn amount (**b**) of tall fescue. Plants were grown for 15 days in solutions under control (0 μM Cd + 15 μM Zn), Cd (75 μM Cd + 15 μM Zn), Zn (0 μM Cd + 115 μM Zn) and Cd + Zn (75 μM Cd + 115 μM Zn) treatments. Error bars represent ± SEM of three replicates. Means with a common letter did not differ significantly (Duncan’s test, *p* < 0.05). The two-way analysis was performed.

**Figure 5 ijms-20-03332-f005:**
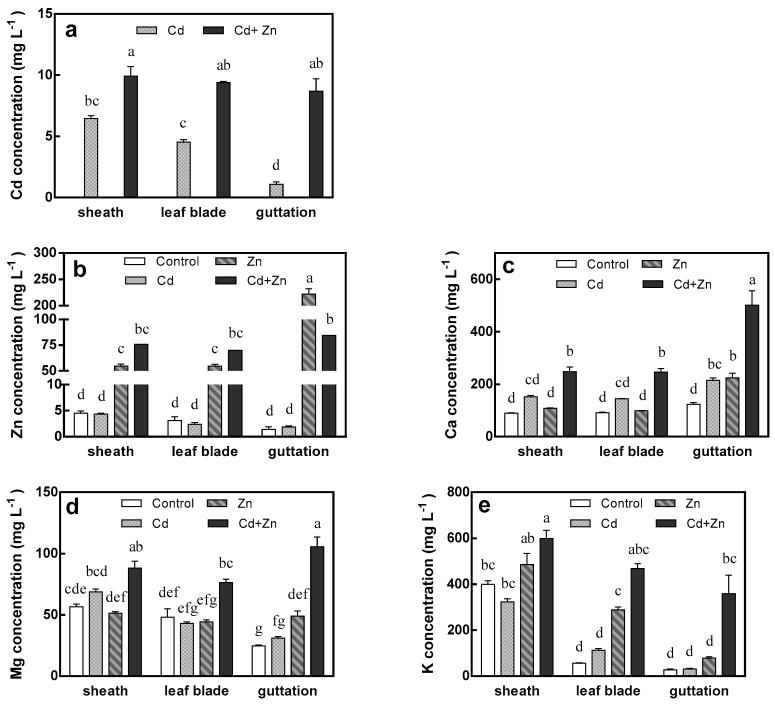
The concentrations of Cd (**a**), Zn (**b**), Ca (**c**), Mg (**d**) and K (**e**) in xylem sap of sheath, leaf blade and guttation of tall fescue. Plants were grown for 15 days in solutions under control (0 μM Cd + 15 μM Zn), Cd (75 μM Cd + 15 μM Zn), Zn (0 μM Cd + 115 μM Zn) and Cd + Zn (75 μM Cd + 115 μM Zn) treatments. Error bars represent ± SEM of three replicates. Means with a common letter did not differ significantly (Duncan’s test, *p* < 0.05). The two-way analysis was performed.

**Table 1 ijms-20-03332-t001:** Significant levels of Cd and Zn treatment effects and their interactions on plant growth and metal concentrations in tall fescue grown for 15 days in solutions under control (0 μM Cd + 15 μM Zn), Cd (75 μM Cd + 15 μM Zn), Zn (0 μM Cd + 115 μM Zn) and Cd + Zn (75 μM Cd + 115 μM Zn) treatments.

Measurements		Cd		Zn		Cd + Zn	
		*F* Value	*p* Value	*F* Value	*p* Value	*F* Value	*p* Value
Plant height		390.97	<0.0001	24.26	0.0004	4.33	0.0596
Shoot biomass		10.84	0.0081	0.13	0.7297	0.68	0.4294
Root length		75.50	<0.0001	0.95	0.3500	0.36	0.5623
Root biomass		34.56	<0.0001	0.07	0.8006	1.46	0.2506
Root Zn concentration		3.68	0.0792	93.93	<0.0001	8.44	0.0132
Shoot Zn concentration		154.56	<0.0001	163.09	<0.0001	36.21	<0.0001
Zn translocation factor		57.98	<0.0001	31.12	0.0001	18.68	0.0010
Zn amount		62.26	<0.0001	132.96	<0.0001	43.12	<0.0001
Zn concentration in xylem sap	Sheath	2.82	0.1318	94.77	<0.0001	2.92	0.1261
	Leaf blade	7.00	0.0294	459.21	<0.0001	8.39	0.0200
	Guttation	37.59	0.0005	280.00	<0.0001	47.74	0.0002
Ca concentration in xylem sap	Sheath	42.26	0.0002	13.40	0.0064	6.06	0.0392
	Leaf blade	38.55	0.0008	16.39	0.0067	8.37	0.0276
	Guttation	28.68	0.0011	26.28	0.0014	8.01	0.0254
Mg concentration in xylem sap	Sheath	17.63	0.0030	1.46	0.2618	4.60	0.0643
	Leaf blade	4.14	0.0762	4.94	0.0569	7.86	0.0231
	Guttation	15.79	0.0041	39.35	0.0002	10.14	0.0129
K concentration in xylem sap	Sheath	0.11	0.7489	10.99	0.0106	2.98	0.1224
	Leaf blade	25.44	0.0015	136.26	<0.0001	6.06	0.0434
	Guttation	2.59	0.1518	7.69	0.0276	3.04	0.1249

The two-way analysis was performed. *F* value was significant when the *p* value < 0.05.

**Table 2 ijms-20-03332-t002:** The concentrations and summations of Cd^2+^ and Zn^2+^ in the roots of tall fescue under different treatments.

Treatment	Cd^2+^	Zn^2+^	Cd^2+^ + Zn^2+^
Control	-	1250.7 ± 101.0 c	1250.7 ± 101.0 b
Cd	4468.9 ± 298.1 a	1774.6 ± 165.5 c	6243.5 ± 452.5 a
Zn	-	7935.9 ± 959.4 a	7935.9 ± 959.4 a
Cd + Zn	2116.2 ± 119.1 b	5375.9 ± 410.4 b	7492.1 ± 450.3 a

The data were presented by means ± standard errors of three replications. Different letters represent the significant differences between the treatments in the same species at LSD 0.05.

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
