# Peer review of "Interaction between Cd and Zn on Metal Accumulation, Translocation and Mineral Nutrition in Tall Fescue (*Festuca arundinacea*)"

_ijms, 2019, doi:10.3390/ijms20133332_

Reviewer 1 Report

The Ms by Dong et al. is  dealing with an investigation aimed at evaluating the effects of a simultaneous supply of Cd and Zn in two concentrations to tall fescue plants under hydroponics. Growth and metal accumulation analyses were mainly performed to assay such effects. The experimental approach was particularly devoted to highlight the interaction of the two metals in affecting such parameters. The topic is certainly of interest as an expansion of knowledge on the exposure of plants to a multi-metal contamination  is certainly requested to better exploit plant the abilities for phytoremediation. Overall, the Ms is characterized by a sound approach and a clear and plain experimental design. I have only a major remark concerning the statistical treatment of data presented in most figures. In fact, in such figures data from different treatments are presented as plotted all together, as clearly evidenced by the letters regarding  the statistical  significance. In this regard, authors did not give info in the M&M section about ANOVA test they performed for evaluating the statistical significance of their data. In my opinion, as the factors of variability affecting their data are more than one (i.e.  Cd concentration and Zn concentration for Figs. 1, 3, 4; metal treatment and plant parts for Fig.5), a multiple way ANOVA should have been run and only whether the interaction of factors (F-value) was significant authors could assign the letters as they did. Contrarily, most of the statements made by authors to present and discuss the data of their work could be not properly corrected.

I have also minor comments listed below:

Title

I think that authors should add “metal” before “accumulation AND translocation” to better highlight that such accumulation and translocation is referred to metals

Keyword

Some items are already present in the title and could be replaced

Results

L 101 I think that the term “addition” is not appropriate, as each treatment was inferred from the beginning of the experiment not as a further step

L113 Please correct the Cd concentration reported

L129 See comment on L101

L 138 Apart from the general consideration on statistics, this statement is not correct as in C the Mg  value is lower in guttation compared to other plant parts

Discussion

L182 I think that the table 1 should be moved to the result section together with the relative data presentation. Moreover, I was not able to find the reference to the Zn and Cd+Zn concentration in Control in the previously reported figures

M&M

The experimental design is sufficiently described but the number of plants used for each replicate is not clear (in L264 is reported as 3 replicates for treatment while in L278 is reported 100 plants per treatment). Please clarify

Conclusion

L304 Basing on the overall data presentation of this work, this statement is not correct as the Cd toxicity observed in all the four biometric parameters (Fig.1) was not alleviated by the Zn treatments, as clearly evidenced by the similar damage observed in both Zn treatments in presence of 75 uM Cd. In any case, it should be taken into account the remark about statistics made in my general comment.

The Ms would benefit from a thorough revision of the language style and punctuation.

I hope my comments will be useful for authors to improve the paper.

Reviewer 2 Report

This is a nice paper reporting results about the “Interaction between Cd and Zn on accumulation, translocation and mineral nutrition in Tall Fescue”. 

The manuscript is pretty well written, pretty clean editorially, presenting a study which seems to be run using sounds laboratory techniques, but the statistical methods need to be improved. This should be run as a 2-way ANOVA.

 Also, it could be sold better: in the introduction the Authors say something about a pre-study but they did not specify anything about it.

I would like to see more “pepper” in the discussion: 

-      what is new? 

-      Why is it novel? 

-      Was it already demonstrated? 

-      In what species? 

-      Why do you only show some minerals and not N and P? 

-      What processes are affected? 

-      Are there any genes involved? 

Since this is the IJMS I would like to see some biosynthetic pathways and gene expression data.

I like this manuscript because so straightforward but maybe this journal is not the right venue for this research.

Author Response

Round  2

Reviewer 1 Report

The revised version of the  Ms by Dong et al. addressed the remarks made to the original submission about the lack of a Multi ANOVA treatment of data reported in some figures, as affected by more than one factor of variability. In fact, authors correctly performed a 2-Way ANOVA where requested, presenting a table (table 1) summarizing the scores obtained under such statistical procedure.  Unfortunately, quite apart from the fact that one statement is  not correct (Cd/Zn interaction is significantly affecting Mg concentration in xylem sap of leaf blade, e.g. P< 0.05) and that no scores were reported for Zn content (Fig. 4b) and  Cd concentration (Fig. 5a), authors did not follow  the results of the statistical assessment to consequently assign the letters of significance. In fact, even if many Cd/Zn interactions are not statistically significant (for example all values regarding Fig. 1 and  some of the data displayed in Fig. 5), authors did not modify the letters of significance accordingly, leaving such letters as reported in the previous version. Therefore, they did not modify the presentation and discussion of such data even though they presented a table (table 1) where it is clearly highlighted that such data are not statistically significant in the way authors presented them.

I have other three minor remarks:

-About M&M, please check the sentence about replicates as it seems that the plants analysed per treatment are 1600 instead of 400 (i.e. 100 plants per replicate *4 replicates * 4 pots each replicate)

-About  conclusion section, in my opinion the sentence (that I remarked in the previous revision) in this form is too speculative and not supported by data provided with this work

-L305 Check and correct the term “phytoremiation”

 Author Response

Reviewer 2 Report

All the suggested revisions were made. The MS is overall improved in quality but the added paragraphs need an extensive English grammar check. 

Author Response

Round  3

Reviewer 1 Report

The new revised version of the Ms Dong et al. is improved by clarifying the concerns raised in previous revisions. Therefore, after a check for language to correct minor errors, it can be accepted for publication.